# Paleoenvironmental Reconstruction for the Last 3500 Years in the Southern Pyrenees from a Peat Bog Core in Clots de Rialba

**Josep-Manel Rodríguez-González** [1,*], **Marc Sánchez-Morales** [1,2,*], **Jordi Nadal-Tersa** [3], **Albert Pèlachs** [3] **and Ramon Pérez-Obiol** [1]

1. GRAMP (Grup de Recerca en Àrees de Muntanya i Paisatge), Unitat de Botànica, Facultat de Biociències, Departament de Biologia Animal, de Biologia Vegetal i d'Ecologia, Edifici C, Facultat de Biociències, Universitat Autònoma de Barcelona, Cerdanyola del Vallès, 08193 Barcelona, Spain

2. Departament de Biociències, Facultat de Ciències, Tecnologia i Enginyeries, Edifici Torre dels Frares, Universitat de Vic-Universitat Central de Catalunya, 08500 Barcelona, Spain

3. GRAMP (Grup de Recerca en Àrees de Muntanya i Paisatge), Departament de Geografia, edifici B, Facultat de Filosofia i Lletres, Universitat Autònoma de Barcelona, Cerdanyola del Vallès, 08193 Barcelona, Spain

* Correspondence: jmrodrig@gmail.com (J.-M.R.-G.); marc.sanchez.morales@uab.cat (M.S.-M.)

**Abstract:** Vegetation landscape dynamics are derived from the relationships established between anthropic activities and climate conditions over time. Paleoenvironmental research in the Pyrenees range (north-eastern Iberian Peninsula) has revealed what these dynamics were like in some regions during the Holocene. However, some fields of biogeography still present questions that need to be addressed, such as the patterns of *Fagus sylvatica* and *Abies alba* and the importance of the fire regime during the Meghalayan (late Holocene). We present a multiproxy study performed in a sedimentary record from the Clots de Rialba peat bog, located at 2093 m a. s. l. (Lleida, southern slope of the Pyrenees mountain range), that covers the last c. 3500 years. Analyses were performed on the organic matter content, pollen, non-pollen palynomorphs, and sedimentary charcoals larger than 150 μm. The palynological spectra revealed a maximum extension of *Abies alba* at about 3500 cal yr BP in the Bronze Age, while *Fagus sylvatica* showed its maximum extent between 3300 and 2800 cal yr BP. A dominance of *Pinus* was detected throughout the studied period. Other taxa such as cereals, herbaceous plants, aquatic plants, and coprophilous fungi have also been discussed to identify anthropic pressure and climate pulses. In addition, the study of sedimentary charcoals reveals the main forest fire episodes and their recurrences, some of them linked to anthropogenic activities and/or climate variations. These anthropogenic activities would include the use of opening and maintaining forest fires in deforestation in order to obtain pastures and spaces dedicated to cereal agriculture and the appearance of some taxa linked to or introduced by human communities.

**Keywords:** Meghalayan; *Abies alba*; *Fagus sylvatica*; Pyrenees; sedimentary charcoals; forest fires

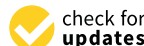



## 1. Introduction

The Pyrenees range, in the northern Iberian Peninsula, is an area of great environmental interest as it constitutes the border of the Eurosiberian and Mediterranean biogeographical regions. Thus, the area has been climatically influenced by both regions over the last millennia. The landscape, however, has not only been modulated by climatic factors, but also by strong anthropogenic pressure, the origin and magnitude of which is still uncertain today. In recent years, several paleoenvironmental studies have shown the importance of sedimentary records from lakes and peatlands on the southern slope of the Pyrenees in explaining plant landscape dynamics [1–11].

As regards the role of climate during the Late Holocene, the Pyrenees has been under the influence of regional variability expressed as millennial-scale oscillations, known as Bond events or cycles [12–18]. Such climate variability could have resulted in cool and wet winters during Bond events 3 to 0 (4.2, 2.7, 1.2 cal kyr BP, and during the Little Ice Age) in the

Western Mediterranean [19]. On the southern slope of the Pyrenees, such climate variability has been characterized by the study of perennial ice deposits in a cave [20], sedimentary records from lakes and peat bogs [4,6,21–28], and dendroclimatological approaches [29]. It is worth highlighting the reviews performed with studies of the area to better determine the evolution of the climate in the region [30,31]. Finally, further away from the Pyrenees but also in the northern Iberian Peninsula, other relevant studies have been carried out based on stalagmite records from the Cantabrian range [14,32], located to the west of the Pyrenees, and from a littoral sequence obtained in Portlligat, to the east of the Pyrenees [33].

Concerning the role that human societies have played in shaping the landscape, archaeological research in Pyrenean high mountain sites has revealed permanent and very ancient occupations up to 10,000 years BP, such as at *Obagues de Ratera rock shelter*, located at 2320 m a. s. l. [34,35]. Other studied sites dating from the Neolithic period are *Sardo Cave* in Boí (1790 m a. s. l.), *Saboredo Great Lake's rock shelter* (2340 m a. s. l.), *Portarró rock shelter* (2285 m a. s. l.), and *Coveta I lake's rock shelter* (2435 m a. s. l.) [36]. Particularly, anthropogenic disturbances in the landscape can be traced back to the Neolithic period [36,37]. These signals include the appearance of new taxa such as cereals tied to the expansion of crops, an increase in forest fires, and booming coprophilous fungi linked to livestock farming [21,38]. Human pressure increased in intensity during the Bronze Age, leaving clear signals within sedimentary records [4,5,8,21,39]. In this sense, discriminating anthropogenic influence from natural events in landscape evolution, even in high mountain sites, has proven to be a challenging exercise, since the climate signal has usually been masked by human activities during most recent times [36].

Some tree species, such as *Abies alba* and *Fagus sylvatica,* can be used to discuss the influence of climate and human activity [2]. In Western Europe, *A. alba* is distributed across the central and southern part of the continent. This species' southwestern border in Europe is in the Iberian Peninsula, with its largest extension of forest area found in the Pyrenees. Its environmental needs include atmospheric and edaphic moisture coupled with dense cover for proper germination and resistance to cold, placing this species below the altitude limits of *Pinus uncinata.* In this respect, *A. alba* can share its ecological range with *Fagus sylvatica.* As regards plant landscape dynamics as a whole, several locations have also been studied on the southern slope of the Pyrenees range [5,8,21,39].

This study aims to provide more data on this issue, and in particular to discuss the role played by forest species such as *Fagus sylvatica* and *Abies alba*, as well as to discern the role of the fire regime at a local level. For this purpose, we present a multiproxy approach performed in a sedimentary record from the Clots de Rialba peat bog (2093 m a. s. l.). We obtained organic matter and palynological and sedimentary charcoal (>150 µm) records spanning the last 3500 years. We sought to characterize the plant landscape, land-uses, and fire dynamics during this period while also providing a more accurate picture of beech and spruce dynamics and the recurrence and intensity of forest fires.

## 2. Study Area

The Clots de Rialba acidic peat bog is an ancient lake of glacial origin currently filled with sediment, located in the axial zone of the Pyrenees mountain range on the edge of the Alt Pirineu Natural Park (42°40′3″ N, 1°1′13″ E, 2093 m a. s. l., Figures 1 and 2). The nearest automatic meteorological station at La Bonaigua (2266 m a. s. l.) records an annual average rainfall of about 1200 mm and temperature averages are below 0 °C from November to April (Figure 3).

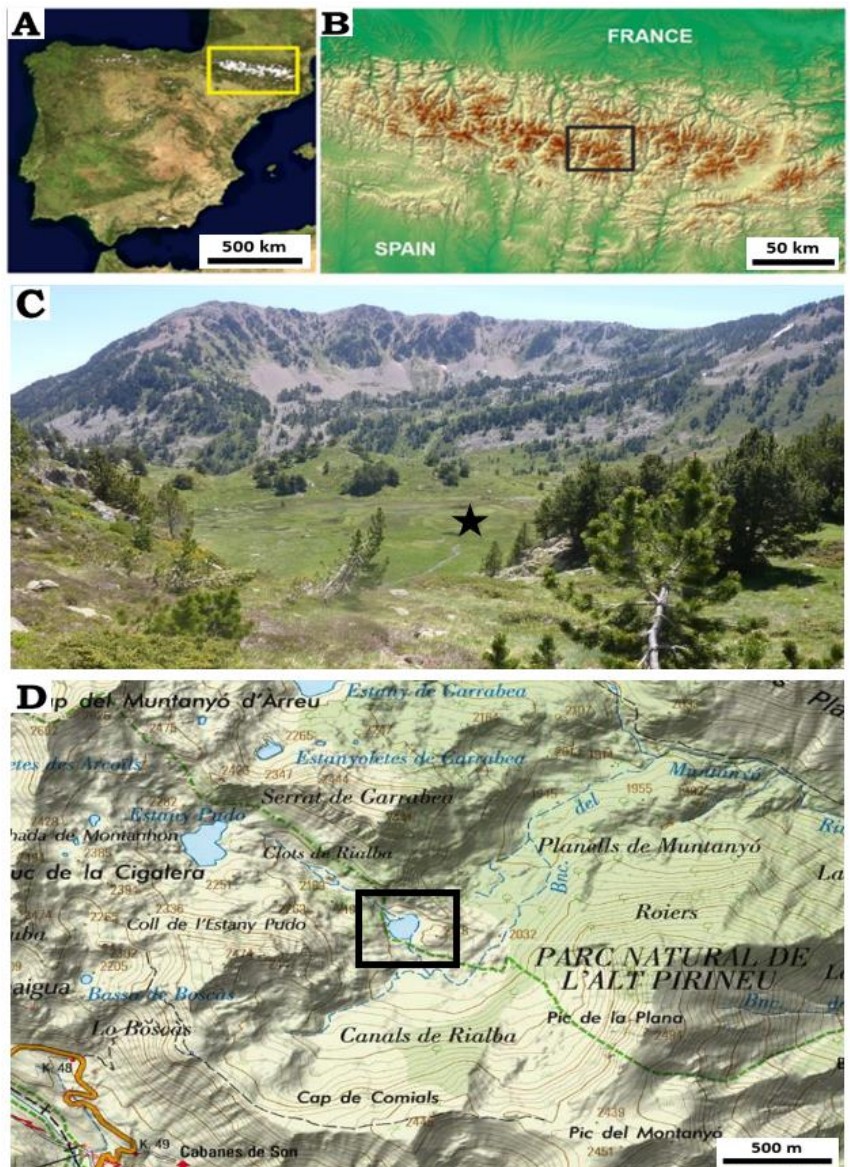

**Figure 1.** Geographical maps and pictures of the Clots de Rialba peat bog. (**A**): Pyrenees range in the northern Iberian Peninsula (yellow square). (**B**): location of the peat bog in the axial Pyrenees range (black square). (**C**): summer scenic view of the Clots de Rialba peat bog with a black star in the sampling point. (**D**): cartographical detail (black square). Source: ICGC.

The outcropping geological materials of the area constitute two major entities: the substratum, consisting of Paleozoic materials ranging in age from Cambro-Ordovician to Devonian periods with rocks of variable composition such as amphibole quartz-gabbro, quartz-diorites, and fine-grained tonalites with amphibole, biotite, and accessory quantities of pyroxenes, and the covering deposits from the Pleistocene and the Holocene (Quaternary). The valley originated from the passage of a glacier and therefore presents a characteristic glacial geomorphology. The bog occupies a basin of glacial origin that has been filled up over time. A link has been established between the geological constitution of the territory and current geodynamics.

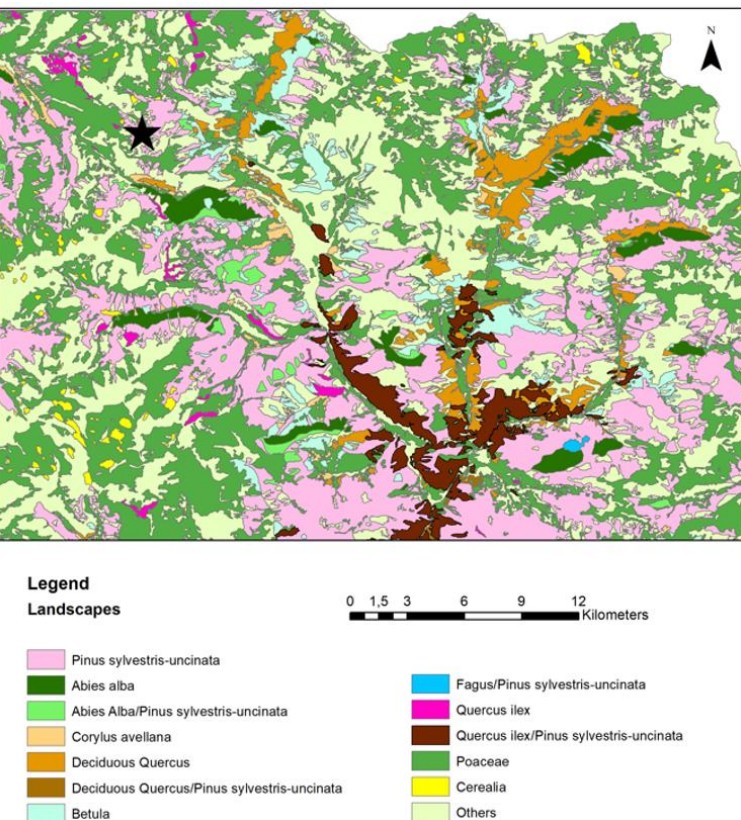

**Figure 2.** Vegetation map of the Clots de Rialba peat bog area. Source: Own elaboration based on cartographic bases of the habitats of Catalonia. Department of Climate Action, Food and Rural Agenda. Catalan Government.

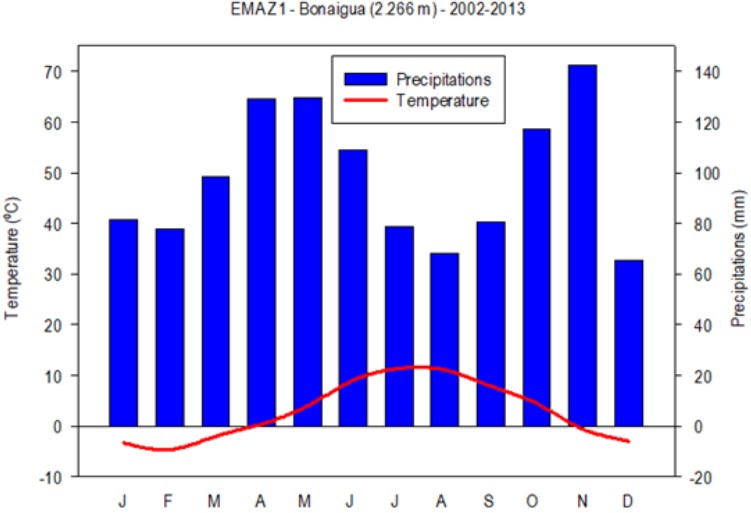

**Figure 3.** Diagram showing temperature and precipitation based on meteorological data from an automatic ground station (Bonaigua, XEMA, 2266 m a. s. l.) very close to the Clots de Rialba peat bog. Source: Catalan Meteorological Service (SMC, meteo.cat).

The vegetation of the surrounding area consists of *Pinus uncinata* forests with *Rhododendron ferrugineum*, acidophilic *Festuca* spp. meadows, and acidic minerotrophic peatlands dominated by *Sphagnum* spp.

The chronology of human occupation in the studied region is complex due to the lack of archaeological sites in this part of the Pyrenees [14,36]. However, recent studies in nearby

areas have shown the growing importance of the agro-livestock system since the Neolithic period, and especially during the Bronze and Roman periods [14,29,37].

Different studies with paleoenvironmental and archaeological data have improved knowledge of the evolution of the landscape on the southern slopes of the Pyrenees [2,4,5,7,15–18,38,39]. This research in Clots de Rialba is the first paleoenvironmental study in this valley contributing to a deeper understanding of this knowledge. The traditional economic system based on agro-livestock and forest activities are in crisis in the mountain areas. This means that there is a replacement of the activities of the primary sector by a service-based economy related to tourism. The decline of pastures and cultivated land causes the forest regeneration.

## 3. Materials and Methods

### 3.1. Samples

During 2013, a survey was conducted in the Clots de Rialba peat bog area with the objective of collecting sedimentary samples for pollen analysis. A continuous sedimentary record (CdR1) was extracted using a PVC tube that measured 10.5 cm in diameter and reached a depth of 225 cm. In the laboratory, the sedimentary record was described and sampled, and a total of 205 samples were obtained for further analysis.

### 3.2. Chronology

To establish an age-depth chronological model for the CdR1 sedimentary record of the Clots de Rialba peat bog, four $^{14}$C AMS dates from peat samples (Table 1) were used and calibrated via the IntCal20 Northern Hemisphere radiocarbon age calibration curve [40]. The age-depth model was then constructed by employing smoothing spline interpolation using the Clam package [40] in the open-source RStudio for R environment [41]. A visual representation of the age-depth model is presented in Figure 4.

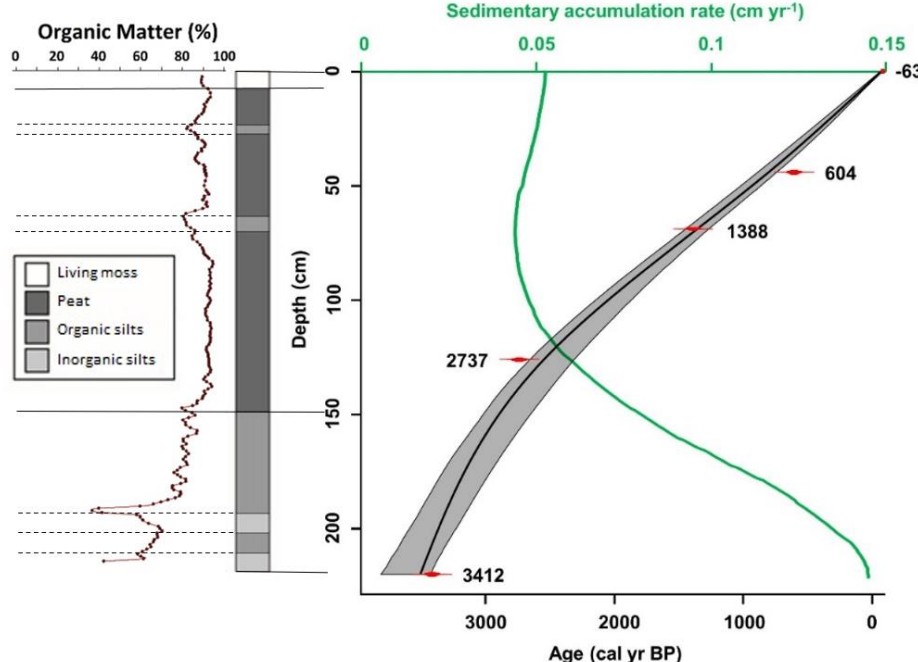

**Figure 4.** Dates, smoothed age-depth chronological model, sedimentary description, organic matter, and sedimentary accumulation rate of the Clots de Rialba record (CdR1).

**Table 1.** Chronological data and calibrated ages for the Clots de Rialba record (CdR1). The IntCal20 Northern Hemisphere radiocarbon age calibration curve was used to calibrate the radiocarbon dates [42].

| Sample Code and Material | Depth (cm) | Lab Code | Conventional Dating BP | $^{13}C/^{12}C$ (‰) | Calibrated Age ($2\sigma$) (cal yr BP) | Median Probability Age, Used for Chronological Model (cal yr BP) |
|---|---|---|---|---|---|---|
| CdR1_M42, Plant material | 44 | Beta-420098 | $600 \pm 30$ | −27.2 | 649–543 | 604 |
| CdR1_M65, Plant material | 69 | Beta-420097 | $1520 \pm 30$ | −25.5 | 1514–1313 | 1388 |
| CdR1_M117, Plant material | 126 | Beta-420096 | $2590 \pm 30$ | −24.8 | 2543–2761 | 2737 |
| CdR1_M205, Plant material | 220 | Beta-371857 | $3190 \pm 30$ | −26.1 | 3455–3364 | 3412 |

### 3.3. Organic Material

The organic matter content of 205 samples from CdR1 was examined by taking 1 g of wet sample for each analysis. The samples were then placed in an oven and dried for 24 h at 60 °C. Once the drying process was complete, the weight of each sample was measured. To determine the organic content, a standard procedure was followed, which involved calculating weight loss after ignition (LOI). The LOI process lasted for 4 h at 550 °C and was performed according to standard procedures [43,44].

### 3.4. Palynology

Pollen analysis was carried out every 2 cm of the CdR1 core (110 samples in total). To calculate the pollen concentration, a tablet containing spores of *Lycopodium* clubmoss was added [45]. Samples were prepared according to standard chemical procedures, including treatments with HCl (10%), HF (70%), KOH (10%), and acetolysis, followed by glycerol mounting [46–48]. The counting and identification were carried out under an optical microscope using reference collections from the Universitat Autònoma de Barcelona (UAB), morphological identification keys [46,47], and visual atlases [49,50].

To determine the pollen percentages, only the total terrestrial pollen was considered, which excludes non-pollen palynomorphs, hygrophyte, and hydrophyte plants. The percentages were then calculated based on this selected pollen. At least 300 pollen grains of terrestrial plants were identified and counted in each sample. The pollen diagram was constructed using Tilia and Tiliagraph software [51,52] and does not include pollen types with low percentage values. Constrained cluster analyses using the incremental sum of squares method (CONISS) were performed to delineate pollen assemblage zones (PAZ) [53].

### 3.5. Sedimentary Charcoals

Sedimentary charcoals (>150 μm) were analyzed on 205 samples from the CdR1 record to determine the fire regime. The chemical procedure followed the classical protocol [54,55]. One $cm^3$ of each sample was soaked with a solution of NaClO (15%) and KOH (5%) [56]. Samples were heated at 70 °C for 90 min with a magnet inside the beaker that moved following an electromagnetic field to facilitate the reaction. When cooled, samples were sieved through a 150 μm mesh under a water jet.

The area of each charcoal fragment was estimated under a stereomicroscope (40× magnification) using an ocular grid with 100 squares. The charcoal counts were first divided by the sample volume and then by the sedimentation rate to obtain two distinct measurements: charcoal concentration (CHAC, particles·$cm^{-3}$) and charcoal accumulation rate (CHAR, particles·$cm^{-2}$·$yr^{-1}$). Charcoal peaks were also identified using a Gaussian

mixture model applied to each 1000-year window, with a threshold of 95% based on the modeled noise distribution.

The signal-to-noise index was >3 throughout the CdR1 record, indicating the suitability of the record for peak detection [57]. To identify the main charcoal peaks and their corresponding magnitudes (particles·cm$^{-2}$·peak$^{-1}$), peak analysis was conducted. All sedimentary charcoal analyses were performed using CharAnalysis v1.1 software [58].

## 4. Results

### 4.1. Sedimentary Description and Organic Matter

Three sedimentary units were distinguished in the Clots de Rialba peat bog (Figure 4). The deepest unit (from 220 cm to 150 cm) was characterized by organic lacustrine silt levels with two narrow phases of inorganic silts, with the lowest organic matter percentage below 40% located in two acute peaks (at 225 cm and 200 cm depth). The inorganic pulses were followed by progressive recovery of organic content along the unit. Above that (from 150 cm to 7 cm depth) the sediment was composed of peat and distinctive thin levels of organic silts (from 70 cm to 63 cm and from 28 cm to 23 cm depth). The upper unit (from 7 cm depth to the top) spanned the last c. 100 years and was characterized by the abundance of hydromorphic soil with living moss on the surface. The organic matter percentage throughout the last two sedimentary units oscillated between 80% and 90%. Each thin band of inorganic silts corresponded to a decrease in organic matter.

### 4.2. Pollen and Sedimentary Charcoals (>150 μm)

The pollen and sedimentary charcoals (>150 μm) diagram from the Clots de Rialba peat bog allowed us to reconstruct the vegetation changes in the studied zone over the last c. 3500 years (Figure 5). Based on the CONISS analysis, five pollen assemblage zones (PAZ) were identified and are described as follows.

### 4.2.1. CdR1-APAZ. from c. 3500 to 2850 cal yr BP

In this PAZ, located between the Middle Bronze Age and the Late Bronze Age, arboreal pollen values ranged between 70% and 90%. *Pinus* was the main arboreal taxon representing between 50% and 70% of the pollen spectra. At the beginning of this period, *Abies* recorded the highest percentage values of the whole sequence, representing 12% of the total pollen spectra. The abundance of fir gradually declined around 3000 cal yr BP, coinciding with the beginning of the Late Bronze Age. *Fagus* showed two peaks over 7% in c. 3450 and 3200 cal yr BP, with posterior regressions and even disappearing in some of the samples. A weak signal of *Olea* was also detected. Cyperaceae was the main aquatic taxon and displayed significant values ranging from 20% to 40%. In terms of anthropogenic indicators, *Rumex* and *Plantago* were detected throughout the period. Cereals were also present (below 1%) and a weak signal of *Secale*, a cereal typical of cold or high-altitude climates and acid soils, was identified around c. 3000 cal yr BP. These cereals were accompanied by pulses of coprophilous fungi (e.g., *Podospora*, *Cercophora*). As regards the pollen accumulation rate, the highest values of the sequence were detected within this period, reaching around 4000 pollen·cm$^{-2}$·yr$^{-1}$ at c. 3500 cal yr BP (Figure 6). These values followed a decreasing trend throughout the PAZ, registering 850 pollen·cm$^{-2}$·yr$^{-1}$ at c. 2850 cal yr BP. Likewise, the pollen accumulation rate of *Pinus*, *Abies*, *Fagus*, and Poaceae also showed a decreasing trend. As for the role of fire, charcoal particles were abundant along this PAZ, exhibiting the most important values recorded throughout the whole sequence (Figure 6). The charcoal accumulation rate displayed values between 0.3 to 0.05 particles·cm$^{-2}$·yr$^{-1}$, while charcoal peaks were also frequent.

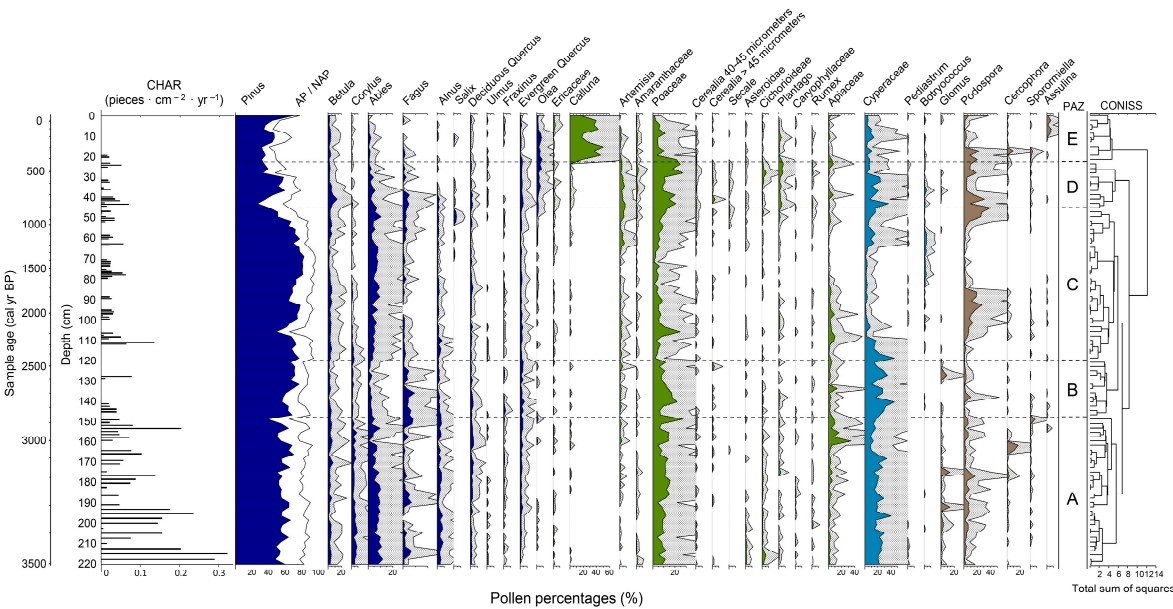

**Figure 5.** Diagram of pollen and non-pollen palynomorphs percentages and charcoal accumulation rate (CHAR) of the Clots de Rialba peat bog (CdR1 record). Light silhouettes indicate 5× exaggeration. Pollen assemblage zones (PAZ) determined by a constrained incremental sum of squares cluster analysis (CONISS) are shown. In dark blue, arboreal taxa; in olive green, shrubs; in blue, hygrophytes and hydrophytes; and in brown, algae and fungi are represented.

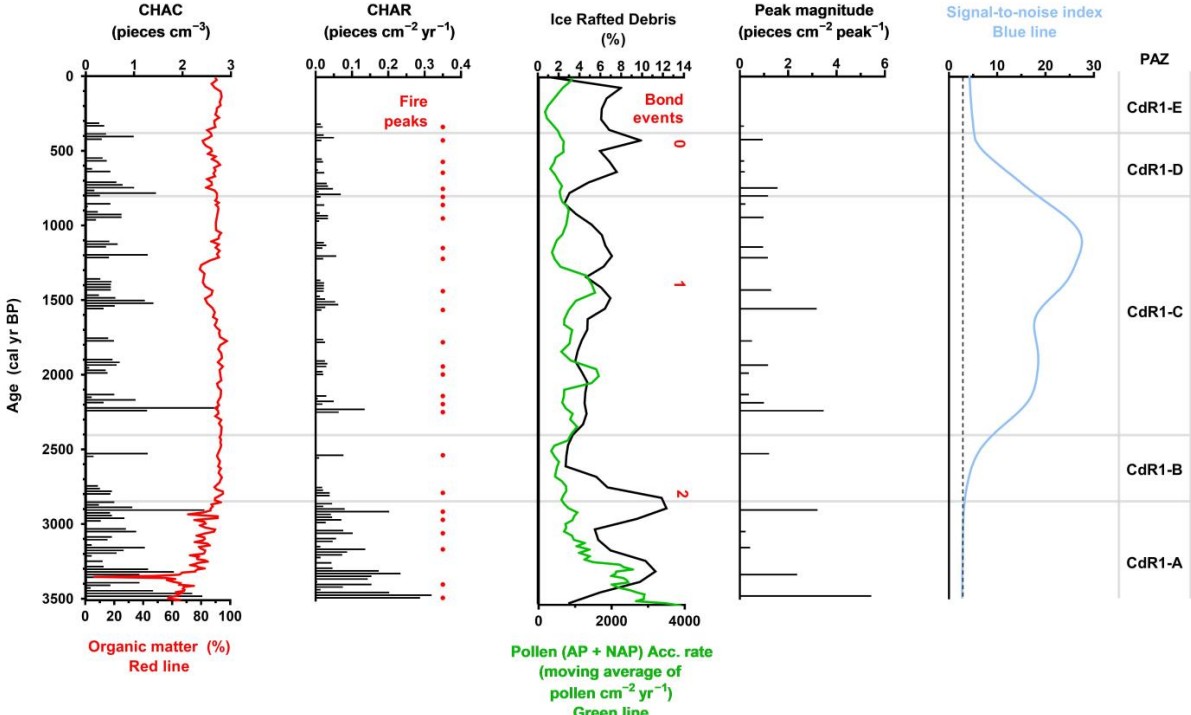

**Figure 6.** Sedimentary charcoals (>150 µm) (CHAC: charcoal concentration; CHAR: charcoal accumulation rate; fire peaks; peak magnitude; signal-to-noise index), organic matter, and pollen (arboreal and non-arboreal) accumulation rate. The Ice Rafted Debris (IRD) index (stack of MC52-V29191 + MC21-GGC22 [13]) is shown for comparison with the CdR1 record. The cutoff value of the signal-to-noise index = 3 [57] is indicated as a black dashed line. Pollen assemblage zones (PAZ) are also shown.

### 4.2.2. CdR1-BPAZ. from c. 2850 to 2500 cal yr BP

In this PAZ, encompassing the Late Bronze Age and the Iron Age, arboreal pollen suffered two steep declines at about c. 3000 and 2800 cal yr BP (<50%), followed by a partial recovery. *Abies* showed the greatest decline in the sequence and its signal even disappeared in some phases. Despite a degree of recovery, the silver fir pollen curve continued to decline until 2700 cal yr BP, at which point the trend reversed and recovered remarkably. Conversely, *Fagus* showed the highest continuous values of the series (5–10%). Apiaceae presented the highest values at the beginning of this phase, cereals maintained their isolated presence, and coprophilous fungi (e.g., *Podospora*, *Cercophora*) displayed several pulses. As a whole, the pollen accumulation rate did not exhibit any significant changes. Fire evidence was still high during this period, albeit rather more spaced over time than in the previous PAZ. Two significant CHAR values (0.2 particles·cm$^{-2}$·yr$^{-1}$) were detected between 2400 and 2300 cal yr BP during the Iron Age.

### 4.2.3. CdR1-CPAZ. from c. 2500 to 800 cal yr BP

After the decline in arboreal pollen recorded at the end of the previous PAZ, the interval from the beginning of Romanization to the end of the Early Middle Ages (EMA) was characterized by progressive recovery of the forest cover, reaching the maximum percentages of the entire sequence that were around 95% in the late Roman period and dominated by *Pinus*. *Abies* also expanded during this PAZ, with values close to 10%, which had not been recorded since the Bronze Age. Likewise, the *Pinus* and *Fagus* accumulation rate additionally displayed two important peaks along this PAZ (Figure 7). As for the role of fire, the charcoal frequency and intensity decreased strongly during this PAZ, with values around 0.1 particles·cm$^{-2}$·yr$^{-1}$, indicating a significant decrease in the fire signal during the Roman settlement. Episodes of low intensity appeared frequently throughout the EMA between c. 1500 and 1200 cal yr BP with a lower intensity than in the Bronze Age.

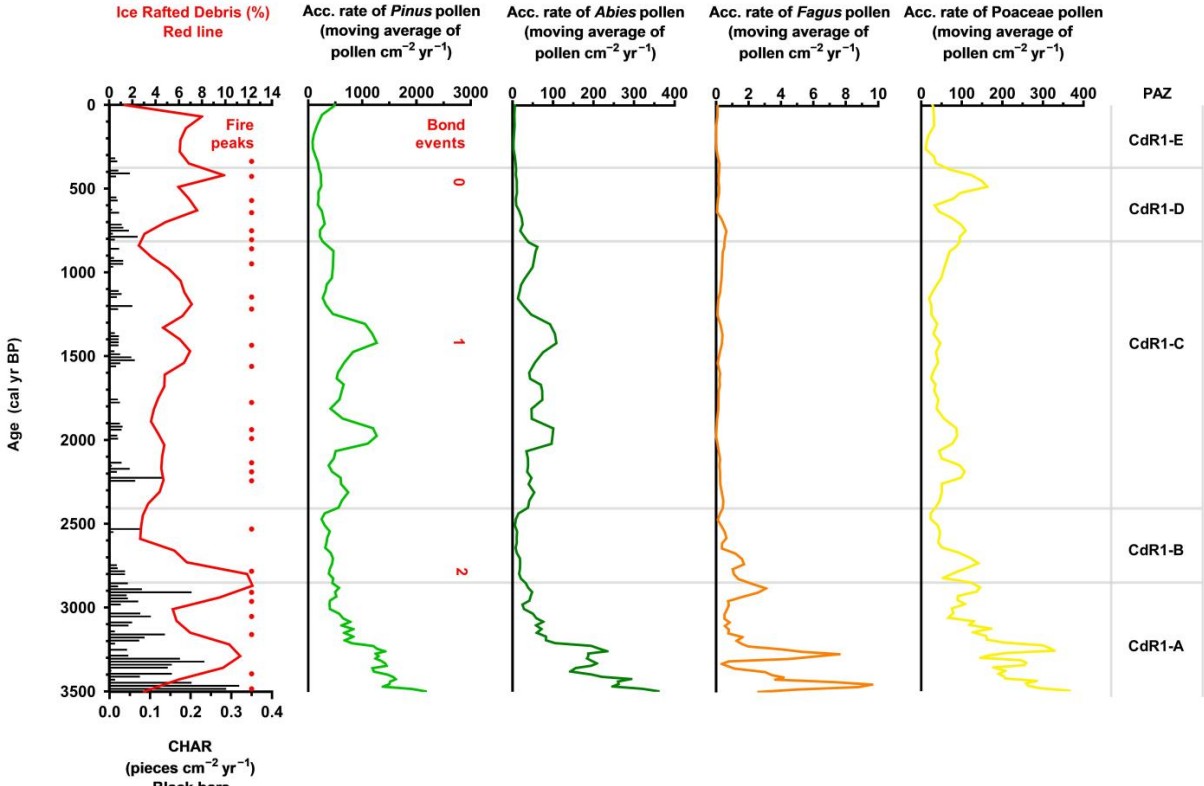

**Figure 7.** Sedimentary macrocharcoals (CHAR and fire peaks), IRD percentage, Bond events, and *Pinus*, *Abies*, *Fagus*, and Poaceae moving average pollen along the PAZ.

#### 4.2.4. CdR1-DPAZ. from c. 800 to 450 cal yr BP

After the late Roman forestry peaks of the previous period, a sharp decline in tree taxa began. In particular, *Pinus* percentages dropped to values below 30%, their lowest for the sequence. *Abies* also registered gentler declines. In contrast, other taxa significantly increased. *Betula* registered the highest values of the whole core at close to 15%. *Corylus* and *Alnus* also increased, although they did not recover the values exhibited during the Bronze Age. Deciduous and evergreen *Quercus* did not seem to be particularly affected by the arboreal decline either, showing modest percentage increases. Mention should be made of *Olea*, an abundantly planted taxon originating from the south-Pyrenean plains and which around 800 cal yr BP experienced a remarkable increase. Since then, it has remained much the same with small ups and downs. In the 16th century, there was an intense rebound of the *Pinus* pollen and consequently the pollen percentages of other taxa rose. All tree pollen fell rapidly again throughout the 17th century CE. Shrubs and herbaceous taxa maintained or increased their percentages, as can be seen in the Ericaceae, *Artemisia*, Poaceae, cereals, *Plantago*, and Apiaceae curves. In particular, the Poaceae accumulation rate registered an important increase during this PAZ (Figure 7). Cereals displayed their maximum expansion coinciding with the major forest decline around the 15th century in the late Middle Ages. It is also worth noting the reappearance of *Secale* in continuous signals from c. 1000 to 300 cal yr BP. Cyperaceae returned to values similar to those recorded in the Bronze and Iron Ages. Coprophilous fungi also increased greatly over the period.

#### 4.2.5. CdR1-EPAZ. from c. 450 cal yr BP to the Present Day

Between c. 450 and 300 cal yr BP, the forest taxa recorded the lowest values of the sequence, extending for almost two centuries. The lowest arboreal percentages in the entire sequence were found in the 18th century CE with values below 50%. In particular, *Abies* and *Fagus* showed a similar regressive trend throughout the period, displaying the minimal accumulation rates of the sequence. *Alnus* and *Corylus* representation also decreased. The presence of cereals was discontinuous and fell with respect to the previous PAZ. *Olea* remained at significantly high values and the high levels of *Plantago* and *Rumex* indicate a clear anthropogenic disturbance and the presence of open spaces. There was a marked decrease in coprophilous fungi such as *Podospora*. During recent decades it is also worth noting intense forest recovery, which unlike other times in the past is mainly due to a resurgence of *Pinus*, which reached values greater than 60% coupled with a certain revival of its accumulation rate (Figure 7). In this PAZ, the fire signal disappeared completely. There was a last faint indication of sedimentary charcoals at about 250 cal yr BP and no further significant signals have been detected since then.

### 5. Discussion

#### 5.1. Landscape Evolution from the Bronze Age to the Iron Age (from c. 3500 to 2200 cal yr BP)

The lack of nearby archaeological evidence has prevented discussion on this issue. However, based on the results obtained (Figures 5–7), we can roughly draw a paleoenvironmental history of the Clots de Rialba peat bog area which complements the data obtained at other sites in the eastern Pyrenees. The CdR1 core spans the last 3500 cal yr BP, from the Middle Bronze Age to the present day. At 3500 cal yr BP, the results revealed a mostly forested area dominated by *Pinus* and accompanied by the highest accumulation rate values of *Abies* and *Fagus* in the whole sequence. *Pinus uncinata* should be the dominant tree species throughout the sequence. The interval between 3500 and 2500 cal yr BP has been characterized as relatively dry in the western Mediterranean [59]. However, it was also inferred that the interval around the Bond 2 event (c. 2700 cal yr BP) was characterized by winter rain maxima due to late Holocene North Atlantic cooling [25]. This climatic scenario characterized by climate variability has provided conditions for the appearance of forest fires in Clots de Rialba, although the anthropogenic factor should not be ruled out either, due to the need to have open areas to favor agriculture, as indicated by the presence of cereals and coprophilous remains from the beginning of the sequence. Fire episodes

were also detected in other sites of the Catalan Pyrenees, such as in the Estanilles and Burg peat bogs [21,38,39]. Now, we provide evidence that these fire events led to changes in the amount of biomass burned, as indicated by the decrease in the pollen accumulation rate (Figure 6).

As regards *Abies* dynamics, this taxon recorded its maximum expansion at 3500 cal yr BP and declined around 3000 cal yr BP. After that point, *Abies* recovered previous values before falling again at between 2800 to 2500 cal yr BP. At other sites in the region, *Abies* also registered a decline until 3000 cal yr BP in Burg Lake (1821 m a. s. l.) before recovering from this time onwards to reach maximum percentage abundances between 2400 and 2000 cal yr BP (up to 15%) [10]. In the Estanilles peat bog (2247 m. a. s. l.), *Abies* showed some parallelism with more or less synchronous expansions and regressions, such as that detected at both sites at around 2400 cal yr BP. Furthermore, in the Bassa Nera peat bog (1900 m a. s. l.) *Abies* presented a maximum at about 4000 cal yr BP and then declined until the present day, showing descending highs and lows in abundance. Overall, there is a similar regression trend for *Abies*, now better understood with the pollen accumulation rate provided by this paper. As previously mentioned, beech and silver fir forests are important components of the region's ecosystem and contribute to the maintenance of ecological balance and diversity. The studied record, furthermore, verifies the post-glacial migration phylogeographical models. It must be considered that Clots de Rialba is located very close to the southern limit of their distribution in the southwest of Europe and that today there are no forests of these taxa nearby the sampling point. The presence of these forests would mean that in the central Pyrenees they were within their suitable climatic area for a certain period (with a wetter and cooler climate) [2,4,5,8,39]. More particularly, the analysis of the *Fagus* dynamics reveals two marked peaks between c. 3500 and 3300 cal yr BP (Figure 7). There may be a link to the double peak of the Bond 2 event, which entailed wetter winters [19], even though human interference is already active. From this moment on, beech tree populations were greatly affected and did not recover previous values. At c. 2900 cal yr BP, strong and sudden decline of the forest cover occurred, coinciding with the fire event. In the Estanilles peat bog, beech appears at around 4200 cal yr BP, associated with colder and wetter climate conditions and the opening of pastures due to anthropogenic causes [39]. Unfortunately, the core recovered in Clots de Rialba only dates back to around 3500 cal yr BP and beech is already present. In both pollen records, differences can be appreciated with respect to the *Fagus* oscillations, which would suggest different local dynamics in the respective valleys.

Overall, human influence seems to prevail in shaping the landscape along the eastern Pyrenees during this time, as suggested by the multiple anthropogenic indicators found at Clots de Rialba and other sites. In Clots de Rialba, the presence of cereals and coprophilous fungi (e.g., *Podospora, Cercophora*) at the start of the sequence (c. 3500 cal yr BP) would indicate anthropogenic interferences which may reflect what happened in earlier stages, such as in the Early Bronze Age. In any case, our findings indicate a strong anthropogenic influence during the Bronze Age, inferred by the presence of *Rumex*, *Plantago*, and a weak but continuous signal of cereals (including *Secale*). Cyperaceae shows significant values, indicating the presence of water available for livestock, and the coprophilous fungi registered different pulses throughout the period, supporting that early livestock activity may have developed. Additionally, in Estanilles, a human intensification phase was identified from 3500 cal yr BP onwards, and the palynological record presents a similar pattern regarding the evolution of the forest cover compared to Clots de Rialba [3,21,39]. Deforestation during the Bronze Age, however, is clearly more intense in Clots de Rialba, a fact also demonstrated by the lower intensity of fires in Estanilles. A great anthropic pressure is also deduced from the Bassa Nera peat bog at c. 3000 cal yr BP. There, forest clearance was interpreted by the intensification of fires, the notable increase in shrubs (Ericaceae), and the rise of agro-livestock indicators [60], a trend also observed in Clots de Rialba.

Concerning the fire regime, the Clots de Rialba fire signals have a similar pattern to those observed in other records in the Pyrenees highlands, such as Estanilles (2200 m a. s. l.) [4,34], but they are in fact weaker than in other lower elevation peat bogs such as Bassa Nera (1900 m a. s. l.) [8,9,60], Burg lake (1850 m a. s. l.) [21] and Llebreta lake (1600 m a. s. l.) [5]. During the Middle and Late Bronze Ages, maintenance fires would have been very frequent in Clots de Rialba, as indicated by the sedimentary charcoal record, helping to increase pasture areas and keeping the forest clearings open. In the Clots de Rialba region, the fires seem to have affected the forest as much as at lower altitudes but were actually designed to burn pastures. Maybe the fire rate was lower because the forest had already burned down beforehand or because initially there was less vegetation as it was close to the treeline.

### 5.2. Landscape Evolution during the Roman Period (from c. 2200 to 1500 cal yr BP)

The rise of the Roman Empire enabled significant recovery of forest cover, including *Abies*. In contrast, no recovery of *Fagus* is detected. In Burg lake, *Fagus* has a low abundance showing a peak during the Roman period. At neither site is there any evidence of forest recovery during the last century [21,38,61]. The results confirm the end of the Roman period as a turning point after *Abies* maximal values were registered. Here the regressive trend of fir populations in the Pyrenees begins and can be observed in a number of places.

Regarding the role of the fire regime, after some centuries with low fire evidence a second intense deforestation episode was recorded in c. 2200 cal yr BP, associated with a fire event. Minor fire peaks were registered, probably indicating that the maintenance of fires was still ongoing. In this sense, new anthropic indicators were registered, such as the presence of *Olea*, which remained low but for the first time persisted throughout the period. Poaceae, on the other hand, declined from 2300 cal yr BP onwards, a pattern also observed in Apiaceae and coprophilous fungi, which dropped strongly and abruptly from c. 1650 cal yr BP onwards. Cereals maintained a weak, discrete, and discontinuous presence throughout this area. A strong increase in *Podospora* might suggest an intensification of livestock activities during the Roman period, followed by a dramatic drop of coprophilous fungi during the Early Middle Ages. Starting from the 2nd century AD, various indicators of Roman civilization began to decline, including urbanization, seaborne trade, and population [62]. The dismemberment of the empire caused local powers to intensively exploit resources that the Romans must have previously controlled. For this reason, the exploitation of iron in the Ferrera valley and the creation of agricultural fields in other valleys increased [5,21,38]. However, it is true that this trend is not necessarily general and may vary according to the resources and populations of each valley [63].

At a local scale, Cyperaceae, which sustained high values throughout the Bronze and Iron Ages, decreased, indicating the existence of an important water column. The presence of wetter local conditions is also supported by the increase in the algae *Botryococcus* (Chlorococcales) that is commonly found in freshwater environments such as ponds, lakes, and rivers, indicating more eutrophic conditions [4].

### 5.3. Landscape Evolution during the Medieval Ages and the Little Ice Age (from c. 1500 cal yr BP to the Present)

In the beginning of the Late Middle Ages, between c. 1000 and 800 cal yr BP, we detected the most abrupt forest mass decline of the whole sequence, mainly due to an increase in the Poaceae pollen accumulation rate and accompanied by other anthropic indicators such as *Plantago*, Cerealia (including a *Secale* signal), and *Rumex*. This scenario of anthropogenic control probably explains the low forest representation in the region during the last millennium. However, we do also detect a pulse of arboreal pollen between c. 800 and 600 cal yr BP, which may be a response to climatic variability. In particular, a shift in conditions was reported in c. 700 cal yr BP in the southern Pyrenees [30]. The warm and relatively dry conditions of the Medieval Climate Anomaly (c. 1100–700 cal yr BP) were followed by the Little Ice Age (from c. 700 to 400 cal yr BP) characterised by wet and cold

conditions, which became more pronounced 400 years ago. In particular, between c. 700 to 200 cal yr BP, a recovery of deciduous taxa and the development of mesophytes were detected in Lake Arreo [23], Montcortès [24], Estanya [25], and Basa de la Mora [26,27], while a decrease in *Quercus suber* and Mediterranean taxa are appreciated in the Portlligat sequence [33]. Thus, there seems to be a climate pulse of regional scope, or at least detected in records from the Pyrenees [30], which could be reflected in Clots de Rialba in the form of the arboreal pollen increase detected between c. 800 and 600 cal yr BP.

As a whole, it is accepted that in the mountain areas around 700 cal yr BP, a prolonged period of cold combined with wetter summers coincided with the widespread famines and plagues that wiped out nearly half of Europe's population in 1347-53 CE [63]. However, in several Pyrenean valleys agro-livestock and mining exploitation were maintained [5,38]. If this happened in the highlands, this depopulation may have led to an abandonment of activities, thereby favoring a certain recovery of the forest cover.

In the most recent period of the sequence, climate oscillations such as the LIA [64] combined with anthropogenic factors, probably due to timber exploitation linked to the fire episodes detected between c. 300 to 200 cal yr BP, could account for the intense forest regression revealed by the pollen spectra. The decline is observed in all tree taxa. In *Abies* and *Fagus* the percentages are also the lowest in the last three millennia. On the other hand, shrub taxa such as Ericaceae and especially *Calluna* showed intense colonization, which would indicate a decrease in humidity in the last three centuries. Another explanation might link *Calluna* with fire events or indicate that it was a result of pasture abandonment at a local level. Cereals again offer weak and discontinuous signals, declining compared to previous centuries, and *Secale* practically disappears from the record as a result of the abandonment of agricultural activities. There is also a marked decrease in coprophilous fungi such as *Podospora*, which suggests a fall in livestock farming intensity compared to earlier phases. Alternatively, it could represent a change in the livestock system and type of cattle or the abandonment of large herds from the second half of the 20th century onwards.

It is also worth noting the intense forest recovery recorded in recent decades, which unlike other times in the past is mainly due to a recovery of *Pinus*. At other sites, renewal of the pine percentages and pollen concentrations could reflect a slight decrease in grazing pressure, particularly in the upper part of the zone [65]. This increase would be clearly linked to an abandonment of agricultural and livestock farming uses. The peat bog is located on the threshold of the *Parc Natural de l'Alt Pirineu*, a protected area since 2003. A last upsurge of coprophilous fungi should also be mentioned, which could be explained by the promotion of extensive cattle ranching in the high mountains around the Natural Park. In fact, cattle feces can currently be identified in the area. In any case, there is still no recovery of fir or beech trees which maintain very low values.

## 6. Conclusions

The results obtained in this study provide a three-and-a-half millennia reconstruction of regional vegetation, forest fires, human impact, and, albeit less clearly, climate dynamics for the central highlands in the Pyrenees. Although it has traditionally been thought that there has not been much human occupation in high mountain areas due to unfavorable conditions, archaeological sites show old and continuous settlements. From the Bronze Ages where our study begins, the anthropic evidence is clear, especially through the use of fire for opening and maintaining pastures.

The pollen and charcoal analyses revealed important disturbances since the Middle Bronze Age affecting the forest cover since all arboreal species declined. Only during Romanization did some arboreal species recover a part of their ancient extension. Particularly significant is the anthropic impact that began in the Medieval period (c. 1000 cal yr BP). It continued until their noticeable recovery over recent decades. Fire episodes were numerous from c. 3500 to 2850 cal yr BP, when the largest number of high intensity episodes over the whole sequence was recorded. Forest clearance and the increase of pasture areas are linked to this epoch. Fire patterns are compatible with forest opening and maintenance clearing

(pasture burning) to preserve agropastoral activities. The fire pattern became less intense over time. After a significant decrease in the fire signal during the Roman occupation, fire signals (sedimentary macrocharcoals) almost disappear in the sedimentary record in the last three centuries. Wildfires probably became maintenance fires in an open landscape and therefore left a smaller signal.

The presence of *Abies* peaked at 3500 cal yr BP and declined significantly over the next millennium. Only in the Roman period, from c. 2100 to 1800 cal yr BP, and later from 1500 to 1200 cal yr BP, did *Abies* relatively recover. After that point, the fir tree again fell to its minimum values of the whole sequence at c. 200 cal yr BP. The presence of *Fagus* peaked between c. 3400 and 3300 cal yr BP and a minor recovery is registered from c. 2900 to 2800 cal yr BP. From this time onwards, evidence of the beech tree in the record has been meager.

Cereal crops do not seem to be especially developed, probably due to the unfavorable climate conditions and despite the detection of weak signals in the Medieval period. However, the continued presence of coprophilous fungi might reflect grazing activities since the middle Bronze Age.

It was difficult to isolate climate signals (e.g., Medieval Warm Period, Little Ice Age, and Bond events) because of the strong and continuous human interference that seems to prevail in shaping the landscape. Nevertheless, Bond event 2 might help to account for some *Fagus* and *Abies* peaks. Additionally, drier local conditions are registered in the most recent centuries with *Calluna* expansion.

**Author Contributions:** All the authors have made substantial contributions to the submission. In particular, J.-M.R.-G. (Conceptualization; Methodology; Investigation; Visualization; Writing—Original Draft), M.S.-M. (Conceptualization; Methodology; Investigation; Visualization; Writing—Review & Editing), J.N.-T. (Conceptualization; Writing—Review & Editing), A.P. (Conceptualization; Methodology; Resources; Writing—Review & Editing; Supervision; Project administration; Funding acquisition), and R.P.-O. (Conceptualization; Methodology; Resources; Writing—Review & Editing; Supervision; Project administration; Funding acquisition). All authors have read and agreed to the published version of the manuscript.

**Funding:** This research was funded by the "Ministerio de Economía y Competitividad" (Spain's Ministry of Economics and Competitiveness, MEC), grant number (CSO2012-39680-C02-02) with the project "Environmental geohistory of fire in the Holocene. Cultural patterns and territorial management since the beginning of livestock and agriculture in the Cantabrian and Pyrenees mountains" and grant number (PID2019-108282GB-I00/AEI/10.13039/501100011033) with the project "Calibration of the indicators of human and climatic influence for the (re)interpretation of postglacial expansion and forest dynamics in the last 18,000 years"; "Estudio biogeografico historico comparado (Montaña Cantabrica, Sistema Central y Pirineos): 18000 años de cambios climaticos y antropicos sobre especies forestales indicadoras" (CSO2015-65216-C2-1-P), awarded to the Department of Geography, Universitat Autonoma de Barcelona, which also awarded an FPI PhD grant (BES-2016-076641) within the project. In addition, the project was funded by the Catalan government's applied geography programme, "Grup de Geografia Aplicada" (AGAUR, Generalitat de Catalunya, 2014 SGR 1090 and 2017 SGR-00343). This project is part of the work of the LTER-Aigüestortes node.

**Institutional Review Board Statement:** Not applicable.

**Informed Consent Statement:** Not applicable.

**Data Availability Statement:** Not applicable.

**Acknowledgments:** The authors wish to acknowledge the special research assistance provided by Miguel Ángel Narváez and Joan Manuel Soriano López and multiple collaborators in the fieldwork: Joan Nunes, M.A. Narvaez, Aaron Pérez-Haase, and Conchy Bueno. We also appreciate the English language review by Stephen Smith. We thank all of them for their help.

**Conflicts of Interest:** The authors declare no conflict of interest.

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
