# Peer review of "Paleoenvironmental Reconstruction for the Last 3500 Years in the Southern Pyrenees from a Peat Bog Core in Clots de Rialba"

_diversity, doi:10.3390/d15030390_

Round 1
Reviewer 1 Report
This paper is well written with good experimental design and profound discussion. I think the conclusion of this paper is reliable, and the paper merits publication after a minor revision. Some minor flaws are as followings:
1. It is unnecessary for the altitude value presented in the title, because it was shown in Materials and Method part.
2. In the end of abstract, the influence of human activities on paleo vegetation should be presented in detail.
3. The authors suggested the environmental changes and fire regimes were mostly triggered by agro-livestock pressure. However, inadequate archaeological evidences were added in the discussion. I wonder an archaeological site distribution map for each phase are needed, and some archaeobotanical and zooarchaeological clues are also needed.
Author Response
Please, see the atachment.

Reviewer 2 Report
In this study, the authors carried out a multiproxy study performed in a sedimentary record from the Clots de Rialba peat bog, that covers the last c. 3500 years. Based on the analysis, the authors revealed that the evolution of regional vegetation, forest fires, and human activities in the region over the past 3500 years. Furthermore, the authors made a hypothesis about the climate dynamics of the region based on the data results. Although this study provides a new reconstruction for understanding the regional paleoenvironment and paleo-human activities, part of the statement and format of the article still have some shortcomings. I provide my comments below and at this stage I would recommend some revisions.
General comments
1. In the introduction part, there should be a systematic review of the previous relevant research rather than a simple list and clarify the significance of this research.
2. Are pollen and fungus percentages calculated separately? If not, it's probably not scientific. The calculation method should be explained in the paper.
3. Abies and Fagus are considered important indicators in the paper. Their ecological significance should be explained in the discussion.
4. In the discussion part, the content should be reorganized and divided into several sections. The discussion should be more logical and clearer.
5. How do you distinguish Secale from Poaceae? Please provide pictures.
6. What is the concentration of pollen in the sediment? May it affect the discussion of the charcoal?
Specific comments
1. Latitude and longitude should be shown on the location map. The location of the core should be marked in the map.
2. The non-English expressions in the text (including the figures) should be changed to English.
3. Supplement the vegetation map of the research area and show the distribution boundaries of Abies and Fagus on the map.
4. When describing pollen percentages, you should add % to each number. eg., ‘70 and 90%’ should be modified to ‘70% and 90%’.
5. Line 300-301: “The influence of climate variability on the landscape” should be unquestionable, not “difficult identify”. Is the so-called “difficult identify” because the results do not reflect the influence of climate change?
6. Line 554: The DOI should be removed to maintain consistency in the format of the references.
7. Figure 1: The figure name and legend obscure some of labels;
8. Figure 4: ‘%’should be marked at the bottom and different species should be classified at the top.
9. Figure 6: The upper axes are misaligned.
Reviewer 3 Report
Overview
The manuscript by Rodríguez-González et alii aims to examine changes in forest composition in relation to human activities, fires, and climate change in a sector of Southern Pyrenees. To this purpose the authors carried out a palynological investigation from Clots de Rialba peat bog sediment, along with a few trusted microcharcoal and statistical analyses relevant to the characterization of landscape evolution and the main factors influencing it.
The methodology is accurately described and suitable to achieve the goals. The original findings are interpreted in light of fire, climatic and human influences. However, the manuscript is not well supported by reference to the regional literature dealing with climatic reconstructions based on proxies independent from pollen. On the whole, the manuscript is logically structured and clearly written, and contains clearly presented and necessary figures. Before the acceptance for publication in Diversity, I would recommend the authors to address the comments/corrections outlined below.
General comments
My major concern regards the inference of local climate variability only using pollen findings interpreted mostly in the light of assumptions from paleoclimate articles not focussed on climate variability of this sector of Spain. I would suggest the authors to improve their discussion by referring to regional palaeoclimate reconstructions from Spain or Western Europe, including those from non-pollen proxies. Recent articles are demonstrating that global climate events can have peculiar regional effects. I think that contextualizing the palynological results in the light this regional palaeoclimate evidence can make this already interesting manuscript much more appealing for researchers dealing with palaeoclimate and palaeoenvironmental changes in Southern Europe. To this purpose I suggest the authors to read the following articles: Sánchez-López et al. 2016, Quaternary Science Reviews 149, 135-150; Baldini et al. 2019, Quaternary Science Reviews 226, 105998; Zielhofer et al. 2019, Climate of the Past 15, 463-475.
The effects of the North Atlantic Oscillation (NAO) variability, which is one of the main factors conveying annual precipitations in this sector of Europe, is not adequately considered in the climatic interpretation of the different intervals of the pollen record. Improving the interpretation also considering this climatic pattern would be desirable.
Technical corrections
Line 97: change “Festucas” to “Festuca”.
Line 184: replace “the majority arboreal taxon” with “the main arboreal taxon”.
Line 197: replace “pollens” with “pollen”.
Lines 234-237: this sentence is difficult to read.
Lines 294-295: “Secale, a high-altitude cereal” is a repetition of lines 192-193 and could be deleted.
Lines 324-327: the discussion on climate variability related to Bond event 2 must be improved by taking into consideration regional references focussed on both pollen and non-pollen paleoclimate proxies.
Lines 327-329: this reference to the Estanilles pollen record must be better discussed otherwise it seems out of context.
Lines 342-344: the interpretation of wetter conditions based on the increase in Botryococcus must be better explained, also because at lines 353-354 an expansion of the same algal palynomorph is related to “more eutrophic conditions”.
Line 357: replace “Starting in” with “Starting from”.
Line 374: write the extended words for LIA abbreviation since it appears for the first time within the text.
Line 374-378: the discussion on Little Ice Age must be improved using the same approach suggest for Bond event 2 (see above comment regarding lines 324-327)
Line 444: write the extended words for MWP abbreviation since it appears for the first time within the text.
Round 2
Reviewer 2 Report
none
Reviewer 3 Report
I appreciated the changes made by Rodríguez-González and co-authors that improved the quality of the interpretations and increased the interest of the whole manuscript for an international audience of scholars dealing with regional specificity of climatic and environmental changes in Southern Europe.